# Green Synthesis and Characterization of CuO Nanoparticles Derived from Papaya Peel Extract for the Photocatalytic Degradation of Palm Oil Mill Effluent (POME)

**You-Kang Phang** [1], **Mohammod Aminuzzaman** [1,*], **Md. Akhtaruzzaman** [2,*], **Ghulam Muhammad** [3], **Sayaka Ogawa** [4], **Akira Watanabe** [4] **and Lai-Hock Tey** [1,*]

1   Department of Chemical Science, Faculty of Science, Universiti Tunku Abdul Rahman (UTAR), Kampar Campus, Jalan Universiti, Kampar 31900, Malaysia; youkang06@hotmail.com
2   Solar Energy Research Institute (SERI), Universiti Kebangsaan Malaysia (UKM), Bangi 43600, Malaysia
3   Department of Computer Engineering, College of Computer and Information Sciences, King Saud University, Riyadh 11543, Saudi Arabia; ghulam@ksu.edu.sa
4   Institute of Multidisciplinary Research for Advanced Materials (IMRAM), Tohoku University, Sendai 980-8577, Japan; s.ogawa@tagen.tohoku.ac.jp (S.O.); akira.watanabe.c6@tohoku.ac.jp (A.W.)
*   Correspondence: mohammoda@utar.edu.my (M.A.); akhtar@ukm.edu.my (M.A.); teylh@utar.edu.my (L.-H.T.)

**Abstract:** In recent years, the green chemistry based-approach for the synthesis of nanoparticles has shown tremendous promise as an alternative to the costly and environmentally unfriendly chemically synthesized nanoparticles. In this study, copper oxide nanoparticles (CuO NPs) were synthesized through a green approach using the water extract of papaya (*Carica papaya* L.) peel biowaste as reducing as well as stabilizing agents, and copper (II) nitrate trihydrate salt as a precursor. The structural properties, crystallinity, purity, morphology, and the chemical composition of as-synthesized CuO NPs were analyzed using different analytical methods. The analytical results revealed that the synthesized CuO was observed as spherical-like in particles with measured sizes ranging from 85–140 nm and has monoclinic crystalline phase with good purity. The Fourier transform infrared (FTIR) spectroscopic results confirmed the formation of the Cu-O bond through the involvement of the potential functional groups of biomolecules in papaya peel extract. Regarding photocatalytic activity, the green-synthesized CuO NPs were employed as a photocatalyst for the degradation of palm oil mill effluent (POME) beneath the ultraviolet (UV) light and results showed 66% degradation of the POME was achieved after 3 h exposure to UV irradiation. The phytotoxicity experiment using mung bean (*Vigna radiata* L.) seed also showed a reduction of toxicity after photodegradation.

**Keywords:** CuO nanoparticles; green synthesis; palm oil mill effluent (POME); papaya peel; biowaste; photocatalyst

## 1. Introduction

Due to the high demand for economical, nutritional, and edible vegetable oil, the palm oil industry has become one of the fast-growing industries in the world. Malaysia has become the second largest palm oil producer in the world after Indonesia since 2006 and the palm oil industry is vigorously contributing to this country's economy. Nevertheless, the palm oil industry is generating a large amount of brownish colloidal liquid waste known as palm oil mill effluent (POME) and previous research studies suggest that 0.5–0.75 tons of POME is generated for every tonnage of fresh fruit bunch processed [1]. POME is typically acidic (pH 4.5–5) with 95–96% water, 4–5% total solids including 2–4% suspended solids, as well as 0.6–0.7% of oil and grease [2]. Moreover, POME also contains organic matters such as lignin (4700 ppm), phenolics (5800 ppm), pectin (3400 ppm), and amino acids [3]. Consequently, POME can cause a severely destructive impact on the environment if discharged directly to water sources owing to its high chemical oxygen

demand (COD) (40,000 to 100,000 ppm) and biochemical oxygen demand (BOD) (25,000 to 65,000 ppm) [4]. In order to keep the aquatic environment safe, the POME must be treated according to the permissible discharge limit set by the Department of Environment (DOE) and Environmental Quality Act (EQA) 1974 of Malaysia's authorities [5]. Various methods and technologies such as aerobic and anaerobic biological treatment, adsorption, solvent extraction, chemical–biological sedimentation, coagulation–flocculation, microalgae treatment, membrane technology, etc. have been adopted in mitigating the polluting effects of POME in aquatic life [2,6–9]. Treatment of POME using advanced oxidation processes (AOP) such as the Fenton process and photocatalysis has gained significant attention and is still developing [10].

In particular, semiconductor nanostructured-based photocatalysis displayed high potential for degradation of various organic pollutants including dyes, surfactants, solvents, pesticides, phenolic compounds, etc. into harmless products under light irradiation. Among nanostructured oxides, copper (II) oxide (CuO) is classified as a semiconductor material type-$p$ with a narrow band gap energy of 1.2 eV and possessing excellent optical, electrical, magnetic, catalytic, and biological properties. Owing to these fascinating properties CuO nanostructures have been reported to be used for a variety of practical applications, such as catalysis, magnetic storage media, biosensor, solar cell, gas sensor, lithium battery, antifungal/anti-microbial agents in the agriculture and health sectors, photocatalysts for wastewater treatment, etc. [11]. Therefore, the synthesis of CuO nanostructures has provoked proficient attention to scientists and researchers. Currently, there are several methods reported in the literature for obtaining CuO nanostructures, such as sol-gel, sonochemical methods, chemical precipitation, hydrothermal, $\gamma$-irradiation, electrochemical reduction, solid-state reaction, and so on [12]. Nevertheless, there are a lot of significant problems associated with these conventional physical and chemical synthesis methods, namely usage of noxious chemicals and solvents, consumption of high energy, generation of toxic waste, high cost, need of special and expensive equipment, along with the tedious procedure. On the contrary, green synthesis of CuO nanostructures using plant extracts has offered a promising choice owing to its simplicity, nontoxicity, low cost, energy efficiency, and environmentally friendliness. The plant extracts contain various phytoconstituents such as alkaloids, terpenoids, flavonoids, polyphenols, sugars, proteins, etc. with a wide range of reductive capacities acting as reducing as well as stabilizing agents during green synthesis. In the past, green synthesis of CuO nanostructures using different plant extracts such as *Bauhinia tomentosa* (leaf) [13], *Euphorbia pulcherrima* (flower) [14], *Madhuca longifolia* (flower and seed) [15], *Lantana camara* (flower) [16], *Capparis spinosa* (leaf) [17], *Aloe vera* (leaf) [18], *Allium cepa,* L. (peel) [19], and banana peel [12] have been demonstrated. In this study, we investigated the use of the water extract of waste papaya peel as reducing and stabilizing agents for the synthesis of copper oxide nanoparticles (CuO NPs).

Papaya (*Carica papaya* L.) is one of the most common fruits distributed throughout the world. Papaya is a rich source of vitamins including $\beta$-carotene, vitamin B (thiamine, riboflavin, niacin and folate), vitamin C, vitamin E, minerals (Na, K, Fe, Ca), and fiber [20]. Papaya juice is consumed for the treatment of various diseases like constipation, dyspepsia, diabetes, cancer, heart stroke, blood pressure, etc. [21]. Owing to high nutritive and medicinal values, a huge amount of papaya is consumed across the world, resulting in the generation of a large amount of waste such as peels and seeds, and this could be a significant source of pollution if these wastes are not being disposed of properly. Utilization of these wastes in various areas could help in mitigating environmental pollution. Keeping this in mind, herein CuO NPs were synthesized in an eco-friendly manner using the aqueous extract of papaya peel waste. The crystallinity, purity, morphology, structural features, and the chemical composition of the green-synthesized CuO NPs were systematically characterized using various analytical tools. The photocatalytic activity of the green-synthesized CuO NPs was also investigated by photodegradation of POME under ultraviolet (UV) irradiation, and photodegradation was monitored by measuring chemical oxygen demand (COD) values. A phytotoxicity test was also conducted with photo-irradiated POME using

mung bean (*Vigna radiata* L.) seed to evaluate residual toxicity and the impact to the aquatic ecosystem.

## 2. Materials and Methods

### 2.1. Materials

Ripened papaya fruits were collected from a local fruit shop in Kampar, Malaysia. Copper (II) nitrate trihydrate [$Cu(NO_3)_2.3H_2O$] was purchased from Quality Reagent Chemical, QRëC, New Zealand, and used without further purification. All glassware was washed with deionized water and dried in an oven before use. The POME sample for this study was directly collected from a local palm oil mill located in the province of Selangor, Malaysia. POME was stored in a black, air-tight container in order to avoid light exposure during transportation.

### 2.2. Preparation of Papaya Peel Extract (PPE)

The ripened papaya peels were washed with deionized water to eliminate dust and organic contaminants. Approximately 100 g of small slices of papaya peels were added to a 500-mL beaker filled with 150 mL of deionized water and heated at 70–80 °C for 20 min. A light yellow-colored solution was formed during the heating period. Upon cooling, the light yellow-colored solution was filtered through vacuum filtration at room temperature. The filtrate (extract) was collected in a 100-mL beaker which was used directly for the synthesis of CuO NPs.

### 2.3. Synthesis of CuO NPs

Scheme 1 shows the synthesis of CuO NPs from copper (II) nitrate trihydrate salt and waste papaya peels. Initially, 40 mL of freshly prepared PPE was heated at 70–80 °C. Then, about 1 g (~0.004 mol) of copper (II) nitrate trihydrate was gradually added into the hot PPE, and the greenish-blue-colored solution was formed immediately. This solution was heated at 70–80 °C with constant stirring as a result the color of the solution gradually changed from greenish-blue to green and the heating was continued until a dark green paste was formed, as shown in Scheme 1. Subsequently, the dark green paste was cooled to room temperature and transferred to a ceramic crucible. Finally, the paste was calcinated in a temperature-controlled muffle furnace at 450 °C for 2 h. After calcination, a fine black-colored powder was obtained, and this was carefully collected andstored in a desiccator at room temperature for characterization using various analytical tools.

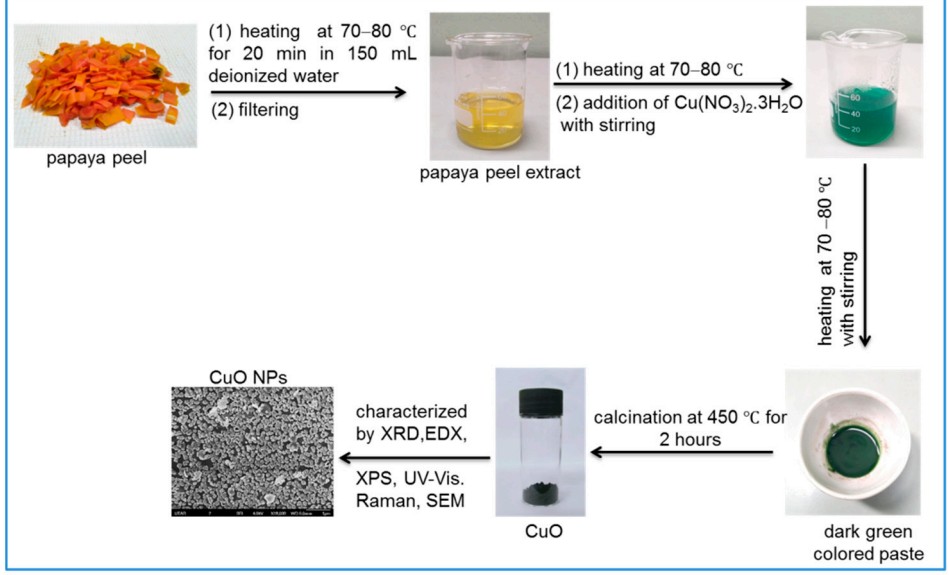

**Scheme 1.** Synthesis of copper oxide nanoparticles (CuO NPs) from waste papaya peels and $Cu(NO_3)_2.3H_2O$ salt.

### 2.4. Characterization

The crystallinity and crystal phase were characterized using a *X*-Ray diffractometer (Shimadzu XRD 6000, Japan) with CuK*α* radiation in the 2*θ* range of 20–80°. A field emission scanning electron microscope (FESEM, JEOL JSM-6710F, Japan combined with EDX (*X*-max, 150 Oxford Instruments)) and a high-resolution transmission electron microscope (HRTEM) (JEOL JEM-3010) were used for morphological and microstructural analysis. UV-vis absorption spectra were recorded using a UV-vis spectrophotometer (Thermo Scientific GENESYS 10S). The FTIR spectra of the CuO sample and PPE extract were recorded by KBr pellet method using a FTIR spectrophotometer (Perkin Elmer RX1) in the region between 4000 to 400 cm$^{-1}$. The *X*-ray photoelectron spectra were obtained using Perkin Elmer PHI5600 (ULVAC-PHI, Inc.). A micro-Raman spectrometer equipped with an optical microscope (Olympus BX51), a CW 532 nm DPSS laser, a Peltier-cooled CCD camera (DV401, Andor Technology), and a monochromator (MS257, Oriel Instruments Co.) were used to measure the Raman spectra.

### 2.5. Photocatalytic Activity of CuO NPs

Prior to investigating the photocatalytic activity of green-synthesized CuO NPs on POME sample, POME was filtered to remove suspended solids using a filtration system reported in our previous study [22]. The photocatalytic activity of the CuO NPs was evaluated by photodegradation of POME under the illumination of an 18 W UV lamp (Roxin, 220–240 V, 50 Hz) in a fully-covered box. In a typical procedure, 150 mg of CuO NPs was added into a beaker containing 300 mL POME and prior to the exposure to UV irradiation, the suspension was magnetically stirred in dark conditions for 30 min to achieve adsorption–desorption equilibrium between POME and photocatalyst. The suspension was then exposed to UV irradiation over 3 h duration while being stirred continuously and was sampled (10 mL) at every 30 min interval for measuring COD using a low-range COD (HACH 21258 vial digestion solution COD-LR) vial from HACH Company, Germany. Then 2 mL of POME (unirradiated and irradiated) was injected into the COD-LR vial which was then digested at 150 °C for 2 h using the HACH DRB 200 COD digital reactor. Subsequently, the COD values of unirradiated and irradiated POME were measured using a Hach UV-vis spectrophotometer DR6000 (Hach, Germany). The efficiency (%) of the COD removal was calculated using the below equation:

$$\% \text{ COD removal} = \frac{COD_0 - COD_t}{COD_0} \times$$

(1)

where $COD_0$ is the initial COD value of the POME (mg/L) and $COD_t$ is the COD value of the POME after photodegradation (mg/L) at different times.

### 2.6. Phytotoxicity Evaluation

The phytotoxicity of POME samples was assessed using mung bean (*Vigna radiata* L.) seed before and after photocatalytic degradation. Prior to the experiment, 0.5 (*w/v*) % NaOCl solution was used to sterilize the mung bean seeds (mung bean seeds with healthy and uniform size were used) and then washed thoroughly with distilled water. Afterwards, cotton pads and mung bean seeds (ten selected seeds) were placed evenly in each sterilized petri dishes (diameter 90 mm). Three different sets of petri dishes were set up and soaked with 5 mL of tap water (control) and 5 mL of POME solutions (before and after photodegradation) at a fixed interval time of 12 h for 7 continuous days at an ambient temperature. After 7 days, the radicle lengths were measured and the phytotoxicity of the POME samples was calculated using the following equation [23]:

$$\text{Phytotoxicity} = \frac{(L_c - L_s)}{L_c} \times 100 \%$$

(2)

where $L_c$ is the radicle length of control and $L_s$ is the radicle length of samples.

## 3. Results and Discussion

### 3.1. Structural Analysis of CuO

The crystal phase and crystallinity of the green-synthesized CuO NPs were studied using XRD. Figure 1 represents the XRD pattern of green-synthesized CuO NPs. The specific diffraction peaks at $2\theta$ = 32.51°, 35.53°, 38.75°, 46.26°, 48.78°, 53.50°, 58.34°, 61.57°, 66.28°, 68.04°, 72.46°, and 75.0° were assigned to (−110), (002), (111), (−112), (−202), (020), (202), (−113), (022), (−220), (311), and (004) lattice planes, respectively, which were well-matched with International Centre for Diffraction Data (ICDD): Entry number-00-045-0937. The diffraction peaks in Figure 1 indicate that the CuO has monoclinic phase with lattice parameters $a$ = 4.6853 Å, $b$ = 3.4257 Å, $c$ = 5.1303 Å, and $\beta$ = 99.549 Å. The green-synthesized CuO NPs were well-crystalline that was confirmed by the well-defined and high intensity diffraction reflection peaks. The crystalline size ($D$) of the CuO NPs was calculated using Debye–Scherrer's equation [24]:

$$D = \frac{0.9\,\lambda}{\beta \cos \theta} \tag{3}$$

where $\lambda$ is the wavelength (CuK$\alpha$), $\beta$ the full-width half maximum (FWHM), and $\theta$ the diffraction angle. The average crystalline size of green-synthesized CuO NPs was found to be 28.06 nm. Raman spectroscopy is a common analysis carried out to analyze the local atomic arrangements and vibrations of the materials in order to study the structural nature of the nanomaterials [25]. The Raman spectrum of the green-synthesized CuO NPs is illustrated in Figure 2. There were three one-phonon modes observed at 289, 323, and 623 cm$^{-1}$, respectively. These peaks can be assigned to the A$_g$ mode of vibration (289 cm$^{-1}$) and B$_g$ mode of vibration (323 and 623 cm$^{-1}$) [26]. In this study, the wavenumbers detected were close to the reported values of monoclinic CuO nanomaterials in previous studies (282, 330, 616 cm$^{-1}$) [27].

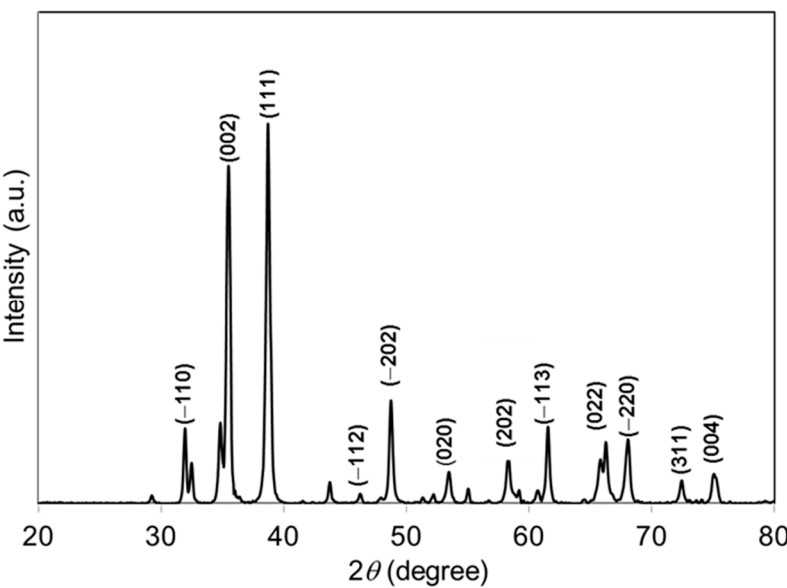

**Figure 1.** XRD pattern of papaya peel-derived CuO NPs.

The chemical composition and purity of the CuO NPs were further investigated by XPS (Figure 3). As shown in Figure 3a, the high-resolution XPS spectrum for the Cu 2$p$ core level shows two peaks at 931.77 and 951.65 eV were identified as the Cu 2$p_{3/2}$ and Cu 2$p_{1/2}$, respectively [28]. There were two peaks located at 941.40 and 961.40 eV higher in binding energy than the main spin-orbit components that corresponded to the shake-up satellites of the Cu$^{2+}$ peaks [29]. Figure 3b shows the O 1$s$ XPS spectra of the sample and according to the O 1$s$ peaks, the O$^{2-}$ in CuO was assigned to the lower energy peak at 528.77 eV while the higher energy peak at 530.52 eV indicated the O that adsorbed on the surface

of the CuO NPs [28,30]. Hence, the XPS measurements excluded the presence of $Cu_2O$ and $Cu(OH)_2$ impurities in the green-synthesized CuO NPs as there would be no satellite peaks observed in Cu $2p_{3/2}$ and Cu $2p_{1/2}$ of $Cu^+$ [31,32]. The appearance of satellite peaks proved to be the presence of the fingerprint of the $d^9$ $Cu^{2+}$ species that were caused by the relaxation phenomenon of the strong configuration interaction in the final state [28]. To further confirm the formation of CuO, EDX analysis was carried out as depicted in Figure 4. The finding indicates the presence strong signals of Cu and O elements in the sample which further revealed the good quality and purity of the green-synthesized CuO NPs.

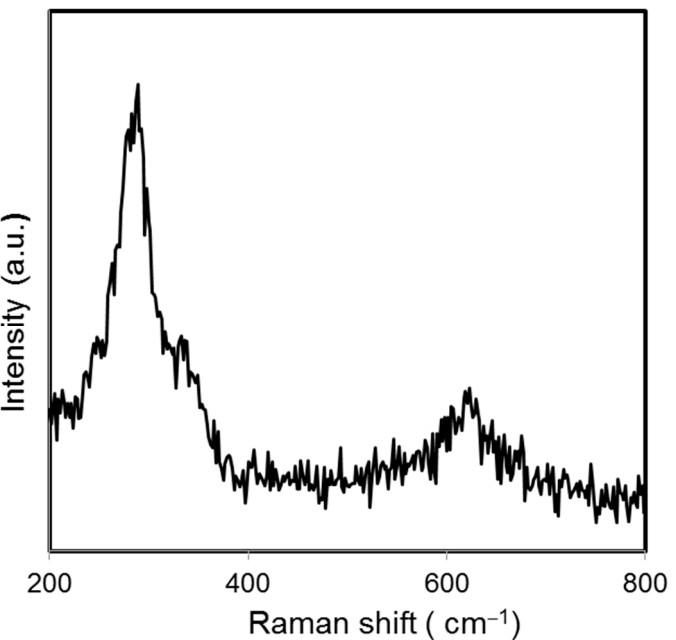

**Figure 2.** Raman spectrum of papaya peel-derived CuO NPs.

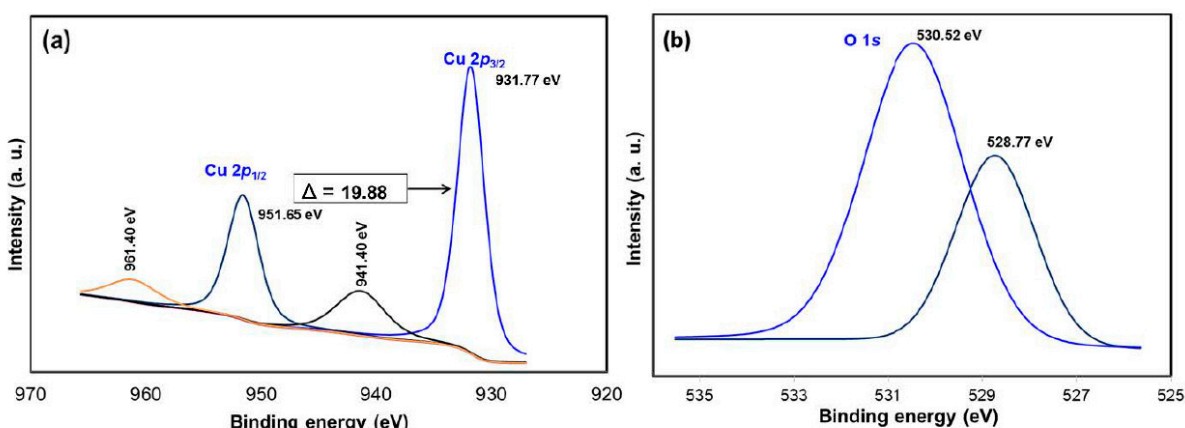

**Figure 3.** XPS spectra of CuO NPs (**a**) Cu *2p* and (**b**) O *1s*.

FTIR spectroscopy analysis was carried out to identify the functional groups involved in the formation of CuO NPs. In order to have a good understanding of the formation of CuO NPs, the papaya peel extract (PPE) was executed for FTIR to compare with that of CuO NPs. Figure 5 shows the spectra for the PPE and CuO NPs. The FTIR spectra of biosynthesized CuO NPs showed a sharp and intense absorption peak at 532 $cm^{-1}$, which was not observed in PPE, expressing the stretching vibration of Cu-O in CuO NPs. The bands at 1412 $cm^{-1}$ for PPE, shifted band at 1384 $cm^{-1}$ for CuO NPs, and peaks at

around 618 cm$^{-1}$ showed the O-H bending of the phenolic group [33]. Stretching of C-O was identified at 1076 cm$^{-1}$ in the spectra of PPE while at a shifted peak of 1120 cm$^{-1}$ for CuO NPs. Absorption bands at 1636 and 1647 cm$^{-1}$ for PPE and CuO NPs, respectively, proved the presence of primary amide groups. The presence of primary amide contributed from the protein content of the *Carica papaya* further proved that protein in the fruit's peel acted as a capping and stabilizing agent in the synthesis of CuO NPs [34]. The absorption bands at 3400 and 3368 cm$^{-1}$ from PPE to CuO NPs indicated the O-H stretching. The C-H stretching observed in PPE (2937 cm$^{-1}$) was not present in CuO NPs. Other weak absorption bands were spotted at 1261, 921, and 778 cm$^{-1}$ in PPE and 833 cm$^{-1}$ in CuO NPs. Finally, natural products in papaya such as phenolic compounds, flavonoids, catechin, etc. could be claimed to be the important chemicals acting as bioreducing agents in synthesizing CuO NPs [35].

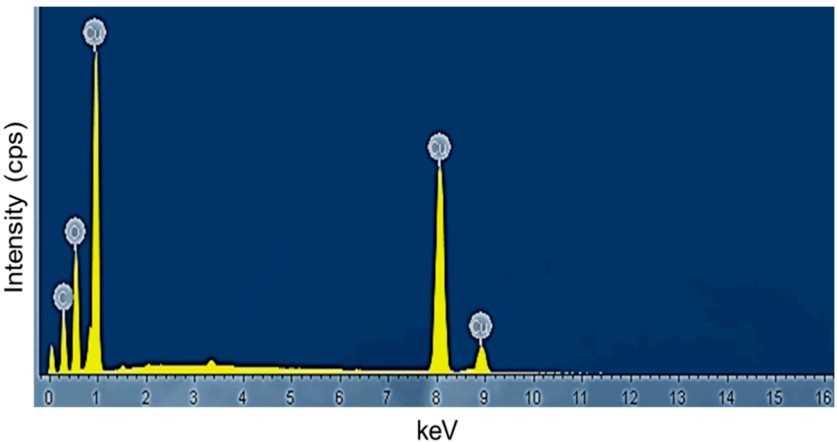

**Figure 4.** EDX spectrum of papaya peel derived CuO NPs.

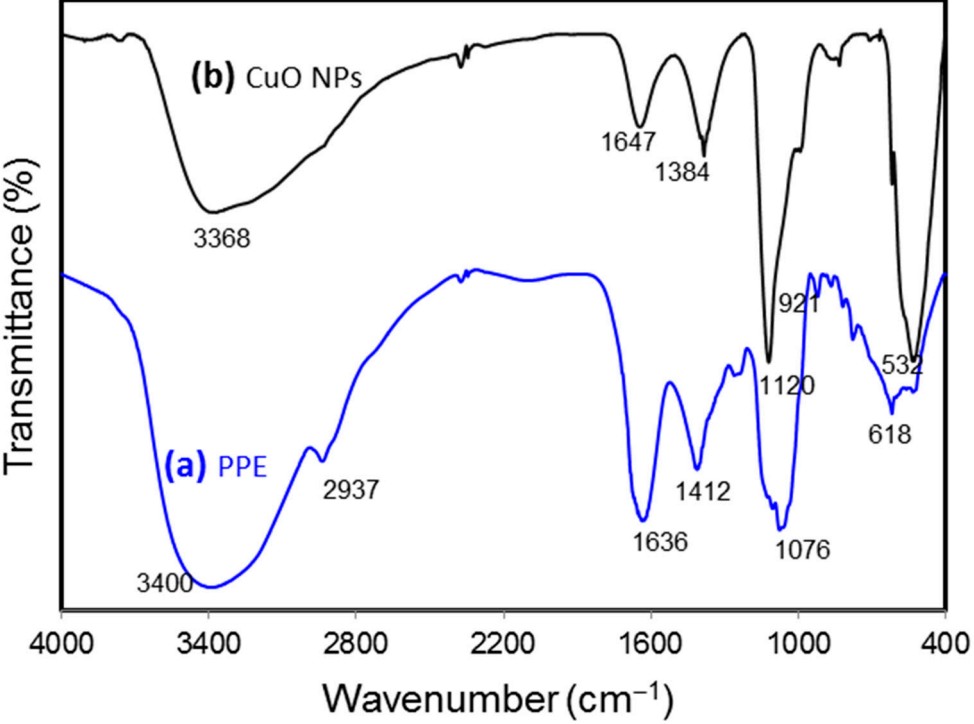

**Figure 5.** FTIR spectra of papaya peel extract (PPE) and green-synthesized CuO NPs.

The detailed morphological and size analyses of green-synthesized CuO NPs were studied using electron microscopy (FESEM and TEM) and the results are shown in Figures 6 and 7. Based on Figure 6a,b, the CuO NPs formed were almost agglomerated spherical in shape with a discrete rough appearance. The particle sizes measured from SEM ranged from 85–140 nm. Figure 7a,b showed papaya peel-derived CuO NPs have different particle sizes and proved that the biosynthesized CuO NPs were densely agglomerated in lump as illustrated by SEM images with a diameter of 277–500 nm. The agglomeration of the nanoparticles was due to the high surface tension [36]. As reported in the literature, CuO NPs have a high tendency to aggregate in ultrapure water during the preparation step before being subjected to TEM measurement [37,38]. Table 1 summarizes the comparison of papaya peel extract-derived CuO NPs with other literatures.

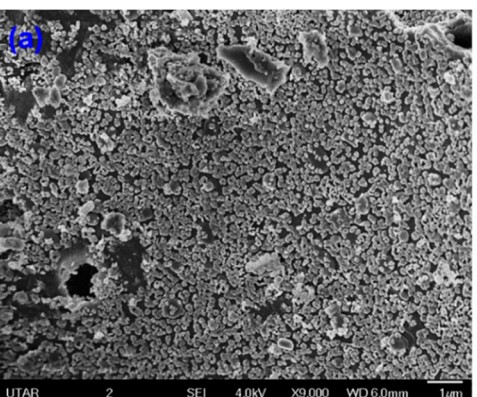 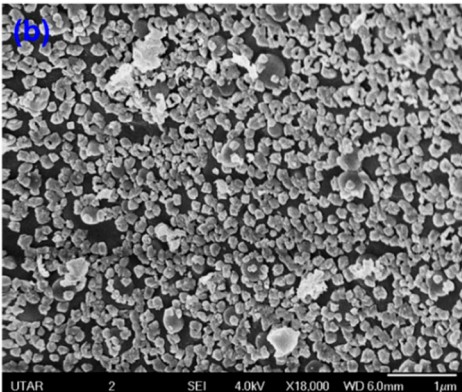

**Figure 6.** SEM images of papaya peel-derived CuO NPs: (**a**) with magnification of ×9000 and (**b**) with magnification of ×18,000.

**Table 1.** Comparison of papaya peel extract-derived CuO NPs with literature.

| Part of Plant Used. | Size (nm) | Shape | References |
|---|---|---|---|
| Flowers (*Anthemis nobilis*) | 18.02–61.29 | Irregular | [39] |
| Peel (*Musa acuminata*) | 50–85 | Spherical | [12] |
| Leaves (*Carica papaya*) | <100 | Square and rectangle | [40] |
| Leaves (*Tinospora cordifolia*) | ~6–8 | Agglomerated spherical | [41] |
| Leaves (*Psidium guajava*) | 19.19 | Elongated spherical | [42] |
| Leaves/stems (*Gundelia tournefortii*) | - | Spherical | [43] |
| Flowers/seeds (*Madhuca longifolia*) | 30–120 | Irregular and spherical | [15] |
| Leaves (*Abutilon indicum*) | 16.78 | Agglomerated irregular and spherical | [44] |
| Seeds (*Caesalpinia bonducella*) | 13.07 | Rice-shaped | [45] |
| Peel (*Carica papaya*) | 85–140 | Agglomerated spherical | Current work |

The UV-vis absorption spectrum of the biosynthesized CuO NPs was recorded at room temperature by dispersing the CuO NPs in deionized water with a concentration of 0.1 wt % (Figure 8). The spectrum showed an absorption peak at 270 nm. This indicated a blue shift occurred when compared to the value of 375 nm reported by Xu and co-workers [46]. Absorption of the CuO NPs at 270 nm resulted from the resonant oscillating of electrons at the conduction band triggered by the incident electromagnetic radiation which is known as surface plasmon resonance [47]. The band gap energy ($E_g$) of the biosynthesized CuO NPs were evaluated by Tauc's plot approach according to the equation [48]:

$$\alpha h\nu = A(h\nu - E_g)^{n/2} \tag{4}$$

where $\alpha$ is the absorption coefficient of the nanoparticles. The $n$ represents the nature of the transition of the electron between the valence and conduction band such that $n = 1$

for the direct band gap while *n* = 4 for the indirect band gap [49]. Symbol A denotes as constant and *hv* defines as the incident photon energy. A plot of $(\alpha h v)^2$ versus *hv* was constructed and the band gap energy for the biosynthesized CuO NPs was estimated through extrapolation of the straight-line part to the *x*-axis which is presented in the inset of Figure 8. The $E_g$ value of the CuO NPs was found to be 3.3 eV in this study. The feature of the plot indicated that it had a direct transition due to the linear absorption at the end of the plot [50]. The higher band gap energy of the CuO NPs than the reported value (1.9–2.1 eV) could be related to the quantum confinement effect that is the band gap increase with a reduction in particle size [51].

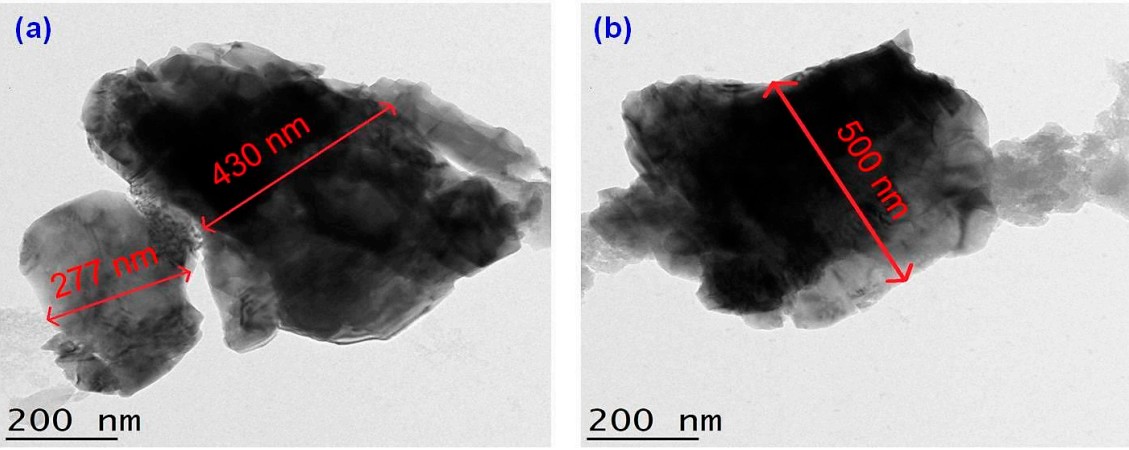

**Figure 7.** TEM images of papaya peel-derived CuO NPs with different particle sizes (**a**,**b**).

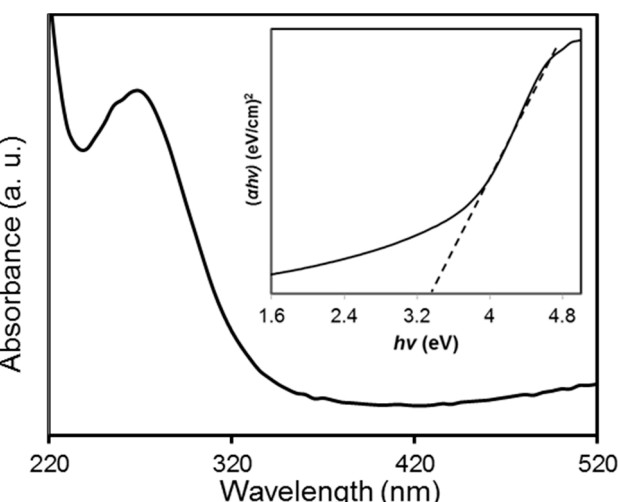

**Figure 8.** UV-vis absorption spectrum of papaya peel-derived CuO NPs at room temperature. The inset shows the plot of the $(\alpha h v)^2$ vs. (*hv*).

### 3.2. Photocatalytic Activity of Green-Synthesized CuO NPs on POME

The heterogeneous photocatalytic treatment of POME was carried out utilizing the biosynthesized CuO NPs as photocatalyst. A blank analysis was also run by exposing the POME sample under direct UV light without adding the photocatalyst. In this study, the blank was used as a control and to prove the potentiality of CuO NPs as a photocatalyst for the degradation of POME. Figure 9 represents the degradation activities of CuO NPs under various experimental conditions. It was noticed that when the POME solution with green-synthesized CuO NPs under dark conditions for 3 h, there was a reduction of

about 10% COD, which could be attributed to the adsorption of organic matter such as carbohydrates, amino acid, lignin, phenolic, etc. within POME on the surface of CuO NPs. In the case of photolysis, the POME solution was irradiated in the absence of CuO NPs, and about 17% COD was reduced after 3 h, indicating the strong resistance of POME solution against UV irradiation. On the other hand, in the presence of CuO NPs, about 66% of COD had been reduced after 3 h UV light irradiation and this reduction in COD was achieved by the degradation of the soluble protein and carbohydrate in POME [52]. The appearance of the POME solution is also shown in the inset of Figure 9. As can be observed, before UV irradiation, the POME had a dark brown color and this POME became pale yellow after exposure to UV light irradiation for 3 h due to photocatalytic degradation of POME.

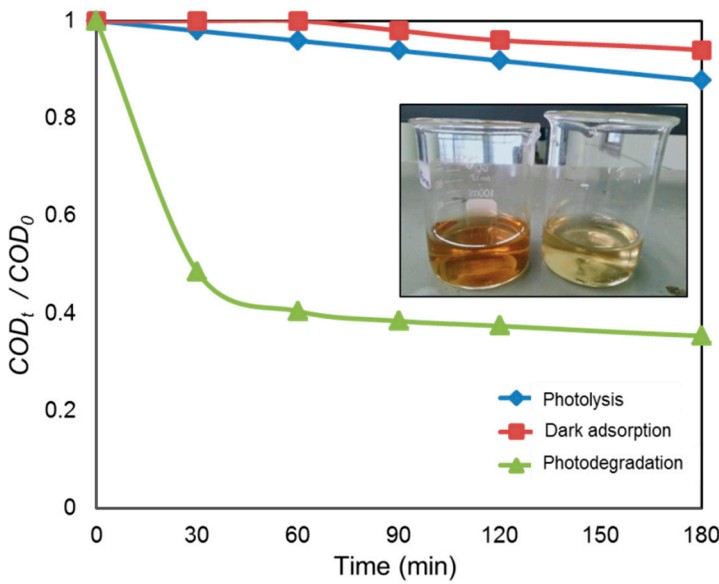

**Figure 9.** Photocatalytic degradation of palm oil mill effluent (POME) using CuO NPs at different experimental conditions. The inset shows the color of POME before and after UV irradiation.

CuO NP is a semiconductor which is able to generate electron-hole pairs ($e^-/h^+$) due to the transition of excited electrons from the valence band to the conduction band, creating holes in the valence band, after absorbing the photon energy ($h\nu$) upon the irradiation of UV light (Equation (5)). The oxygen and water molecules in the POME reacted with the electron-hole pairs and subsequently formed hydroxyl radical ($OH^\bullet$), which was responsible for breaking down the organic complex in POME. The oxygen ($O_2$) reacted with the excited electrons ($e^-$) and dissociated hydrogen ions ($H^+$) from water molecules to form hydrogen peroxide ($H_2O_2$) (Equation (6)). Under the irradiation of UV, the hydrogen peroxide was generated, which further reduced to hydroxyl radicals and hydroxide ions (Equation (7)). The hydroxyl radicals were involved in degrading the organic materials in the POME into simple harmless degradation products. In the end, the possible products of the photocatalytic degradation process could be water, carbon dioxide, methane gas, and other simple molecules (Equation (8)) [52]. A similar mechanism has been proposed for the photocatalytic degradation of organic pollutants in aqueous media using CuO nanomaterials [53,54]. The schematic diagram of POME photocatalytic degradation mechanism over green-synthesized CuO NPs is illustrated in Scheme 2.

$$\text{CuO NPs} + h\nu \text{ (UV light)} \rightarrow \text{CuO NPs } (e^- + h^+) \tag{5}$$

$$2e^- + O_2 + 2H^+ \rightarrow H_2O_2 \tag{6}$$

$$e^- + H_2O_2 \rightarrow OH^\bullet + OH^- \tag{7}$$

$$OH^\bullet + \text{POME} \rightarrow \text{intermediates} \rightarrow H_2O + CO_2 + CH_4 \tag{8}$$

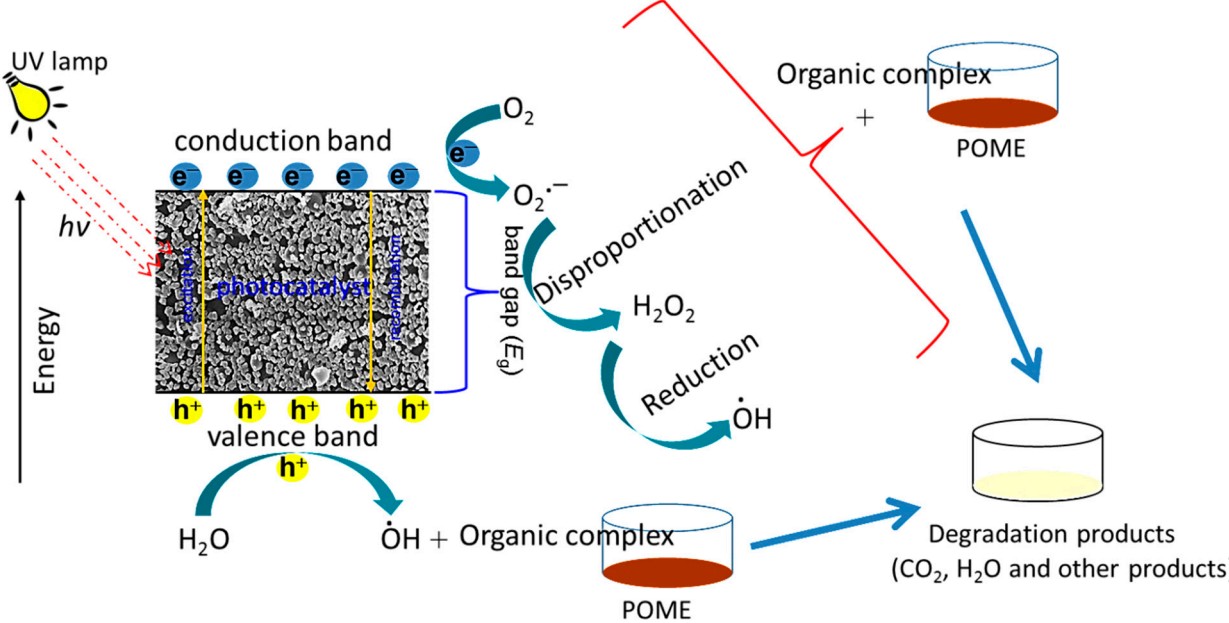

**Scheme 2.** Schematic diagram of POME photocatalytic degradation mechanism using papaya peel-derived CuO NPs.

As the photo-treated POME was going to be discharged into water resources at the end, hence, phytotoxicity test using mung bean seed was performed as shown in Figure 10 to evaluate the impact on the aquatic ecosystem.

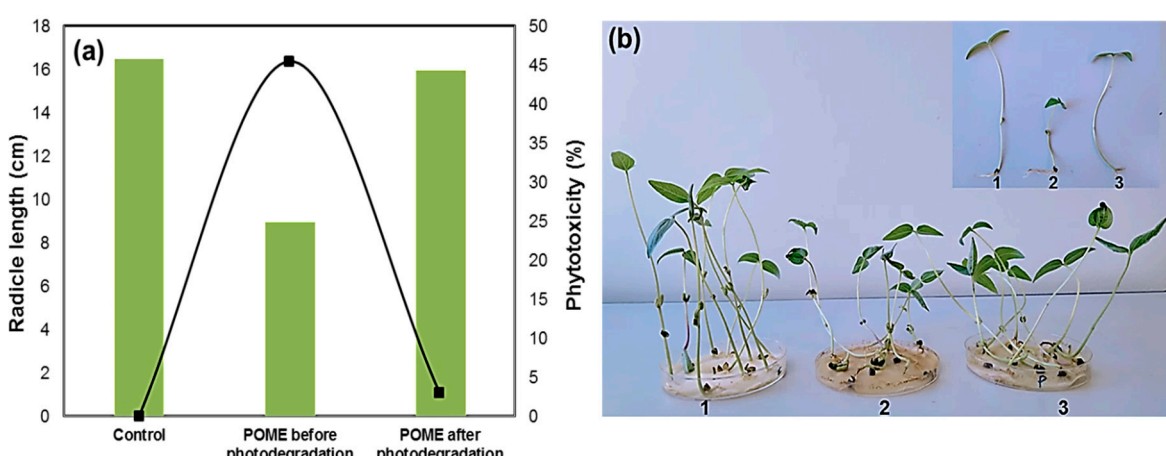

**Figure 10.** (**a**) Phytotoxicity of POME before and after photodegradation over green- synthesized CuO NPs; (**b**) photograph showing mung bean (*Vigna radiata* L.) seed germination in the sample: 1. control, 2. before degradation, 3. after degradation.

Compared with tap water (control), the untreated POME solution strongly inhibited the germination of mung bean seed, where the initial was phytotoxicity 45.5%. Interestingly, the phytotoxicity level reduced significantly to 3.0% when mung bean seed was exposed to the photo-treated POME solution. The radicle length in mung bean also indicated the reduction in phytotoxicity. The average radicle length was found to be 9.0 cm for the untreated POME solution, but the radicle length was greater when the seed was grown in photo-treated POME solution (16.0 cm) as well as tap water (16.5 cm). Therefore, these results indicated that green-synthesized CuO NPs have good photocatalytic activity to reduce phytotoxicity of industrial wastewater. The organic pollutant degradation efficiencies of the CuO photocatalyst are compared in Table 2.

**Table 2.** Comparison of photocatalytic performance of CuO nanomaterials on the degradation of organic pollutants.

| Precursor | Synthesis Pathway | Pollutants | Irradiation Source | Time (min) | Efficiency (%) | References |
|---|---|---|---|---|---|---|
| $CuSO_4 \cdot 5H_2O$ | Biologically | 4-Nitrophenol | UV light | 20 | 99.5 | [55] |
| $CuSO_4 \cdot 5H_2O$ | Biologically | Crystal violet | Visible light | 300 | 97 | [56] |
| $Cu(NO_3)_2$ | Chemically | Rhodamine B | Visible light | 180 | 97.6 | [57] |
| $Cu(NO_3)_2 \cdot 3H_2O$ | Biologically | Bromothymol blue | Sunlight | 180 | 100 | [58] |
| $CuCl_2 \cdot 2H_2O$ | Biologically | Methylene blue | Sunlight | 120 | 99.3 | [59] |
| $CuSO_4 \cdot 5H_2O$ | Biologically | Vat Red 13 | UV light | 60 | 90 | [60] |
| $Cu(NO_3)_2 \cdot 3H_2O$ | Biologically | POME | UV light | 180 | 66 | Current work |

## 4. Conclusions

In summary, a green and cost-effective method for the synthesis of CuO NPs using the waste papaya peel extract which focused on the utilization of waste was demonstrated. We synthesized CuO NPs using the nontoxic and renewable aqueous extract of waste papaya peel and copper (II) nitrate trihydrate salt as a precursor. Waste papaya peel extract-mediated synthesized CuO NPs are pure, crystalline with band gap energy of 3.3 eV. The photocatalytic activity of the green-synthesized has been assessed by the photodegradation of POME. The results revealed that CuO NPs have a significant photocatalytic performance in degrading POME with reduced phytotoxicity and can thus be used as a promising photocatalyst in POME wastewater treatment.

**Author Contributions:** Methodology, investigation, and writing—original draft preparation, Y.-K.P.; conceptualization, writing—review and editing, M.A. (Mohammod Aminuzzaman); writing—review and editing, M.A. (Md. Akhtaruzzaman); funding acquisition, writing—review and editing, G.M.; data curation, S.O.; writing—review and editing, A.W.; conceptualization, supervision, funding acquisition, L.-H.T. All authors have read and agreed to the published version of the manuscript.

**Funding:** This research was funded by UNIVERSITI TUNKU ABDUL RAHMAN (UTAR), grant number: IPSR/RMC/UTARRF/2018-C2/T03. The authors are thankful to Researchers Supporting Project (RSP-2020/34), King Saud University, Riyadh, Saudi Arabia. The authors also acknowledge Universiti Kebangsaan Malaysia (UKM) for funding this study with the grant number: GP-2020-K020067.

**Institutional Review Board Statement:** Not applicable.

**Informed Consent Statement:** Not applicable.

**Data Availability Statement:** The data presented in this study are available on request from the corresponding authors.

**Acknowledgments:** We would like to gratefully acknowledge Universiti Tunku Abdul Rahman (UTAR) for funding the study with the grant number of IPSR/RMC/UTARRF/2018-C2/T03 and providing the necessary facilities throughout the research. The authors are thankful to Researchers Supporting Project (RSP-2020/34), King Saud University, Riyadh, Saudi Arabia. The authors also acknowledge Universiti Kebangsaan Malaysia (UKM) for funding this study with the grant number: GP-2020-K020067.

**Conflicts of Interest:** The authors declare no conflict of interest.

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
