# Peer review of "Green Synthesis and Characterization of CuO Nanoparticles Derived from Papaya Peel Extract for the Photocatalytic Degradation of Palm Oil Mill Effluent (POME)"

_sustainability, doi:10.3390/su13020796_

Round 1

Reviewer 1 Report

In this paper, the authors describe the preparation of CuO particles by a green method using papaya peel extract. The resulting articles are well characterized and they were tested successfully in photocatalytic degradation of palm oil mill effluent (POME). The results are interesting and the catalytic activity of the CuO prepared in this work is very good. This reviewer recommends adding analyses of the particles after using in photocatalytic degradation of POME.

Author Response

Thank you very much for your valuable comments on our manuscript. We would like to answer your comments as follows;

  1. Stability and reusability is a serious issue while developing any photocatalyst for waste water treatment as it will affect the cost of the treatment process. Currently, we are examining reusability of our green -synthesized CuO NPs photocatalyst in the degradation of POME for different cycles and to evaluate its performance for each cycle. We are also analyzing CuO NPs using XRD, SEM, EDX  and FTIR  after using photocatalytic degradation of POME  and doing surface treatment of the used green-synthesized CuO NPs to enhance the degradation efficiency of POME, will be reported in future.

Reviewer 2 Report

Manuscript: “Green synthesis and characterization of CuO nanoparticles derived from
papaya peel extract for the photocatalytic degradation of palm oil mill
effluent (POME)”

This manuscript reports a green bio waste synthesis of CuO nanoparticles derived from papaya peels extract applied to photocatalytic degradation of palm oil mill effluent.

Several techniques such as XRD, SEM , FTIR, XPS, TEM, Raman spectroscopy, EDX, UV-Vis absorption, Photocatalytic degradation of POME using CuO NPs are used for the experimental investigations.
The results showed a successful synthesis of CuO NPs by using a mix of copper(II) nitrate trihydrate salt [Cu(NO3)2.3H2O] and waste papaya peels and a good efficiency of these CuO NPs for photocatalytic activity for degradation of palm oil mill effluent (POME) beneath the ultraviolet (UV) light.

I find this paper interesting as well as well written, It deserves consideration for publication in the Sustainability Journal by MDPI.
I have only two points, which can be addressed easily by authors:

1) Can authors add a sentence on the sample CuO NPs preparation, if the copper(II) nitrate trihydrate salt was not present in the solution with papaya peels extract?

2) Due to a similar green frame, can authors add a reference of recently published paper on green CuO nanostructures present in C pyrolyzed power obtained from Alliu m Cepa (L..) peels (Appl. Sci. 2020, 10, 3819; doi:10.3390/app10113819)?

Author Response

Thank you very much for your valuable comments on our manuscript. We would like to answer your comments as follows;

  1. We tried to synthesize CuO NPs using Scheme1 without adding the copper(II) nitrate trihydrate salt unfortunately unable to obtain CuO NPs. This suggesting copper (II) nitrate trihydrate salt must be added in papaya peel extract as a source of       Cu2+ ion for the synthesis of CuO NPs through the bioreduction process which involving various phytochemicals in the papaya peel extract.
  2. The reference recommended by Reviewer 2 was added into the manuscript at the Reference number [19] as listed in the reference section.